# A Novel Class of Injectable Bioceramics That Glue Tissues and Biomaterials

**DOI:** 10.3390/ma11122492

**Published:** 2018-12-07

**Authors:** Michael Pujari-Palmer, Hua Guo, David Wenner, Hélène Autefage, Christopher D. Spicer, Molly M. Stevens, Omar Omar, Peter Thomsen, Mattias Edén, Gerard Insley, Philip Procter, Hakan Engqvist

**Affiliations:** 1Applied Material Science, Department of Engineering, Uppsala University, 75121 Uppsala, Sweden; michael.palmer@angstrom.uu.se (M.P.-P.); david.wenner@angstrom.uu.se (D.W.); gerard@gpbio.net (G.I.); philip.procter@angstrom.uu.se (P.P.); 2Department of Materials and Environmental Chemistry, Stockholm University, 10691 Stockholm, Sweden; hua.guo@mmk.su.se (H.G.); mattias.eden@mmk.su.se (M.E.); 3Department of Medical Biochemistry and Biophysics, Karolinska Institute, 17177 Stockholm, Sweden; helene.autefage@ki.se (H.A.); cspicer20@gmail.com (C.D.S.); 4Department of Materials, Department of Bioengineering and Institute of Biomedical Engineering, Imperial College London, London SW7 2AZ, UK; m.stevens@imperial.ac.uk; 5Department of Biomaterials, Institute of Clinical Sciences, University of Gothenburg, 40530 Gothenburg, Sweden; omar.omar@biomaterials.gu.se (O.O.); peter.thomsen@biomaterials.gu.se (P.T.)

**Keywords:** cement, tissue adhesive, phosphoserine, self-assembly, amorphous calcium phosphate (ACP), correlation nuclear magnetic resonance (NMR) spectroscopy, bioinspired, biomaterial

## Abstract

Calcium phosphate cements (CPCs) are clinically effective void fillers that are capable of bridging calcified tissue defects and facilitating regeneration. However, CPCs are completely synthetic/inorganic, unlike the calcium phosphate that is found in calcified tissues, and they lack an architectural organization, controlled assembly mechanisms, and have moderate biomechanical strength, which limits their clinical effectiveness. Herein, we describe a new class of bioinspired CPCs that can glue tissues together and bond tissues to metallic and polymeric biomaterials. Surprisingly, alpha tricalcium phosphate cements that are modified with simple phosphorylated amino acid monomers of phosphoserine (PM-CPCs) bond tissues up to 40-fold stronger (2.5–4 MPa) than commercial cyanoacrylates (0.1 MPa), and 100-fold stronger than surgical fibrin glue (0.04 MPa), when cured in wet-field conditions. In addition to adhesion, phosphoserine creates other novel properties in bioceramics, including a nanoscale organic/inorganic composite microstructure, and templating of nanoscale amorphous calcium phosphate nucleation. PM-CPCs are made of the biocompatible precursors calcium, phosphate, and amino acid, and these represent the first amorphous nano-ceramic composites that are stable in liquids.

## 1. Introduction

Calcium phosphate cements (CPCs) are effective bone void fillers, due, in part, to their chemical similarity to the inorganic phase of bone (hydroxyapatite), which facilitates the deposition of new extracellular matrix, tissue, and vasculature on the bioceramic surface (osteoconduction). In the mineral phase of calcified tissues, the proteinacious organic phase creates complex material and biological properties that are absent from synthetic bioceramics: the hierarchical organization of organic and inorganic phases into a nanoscale composite, biologically controlled mineralization (biomineralization), and tissue integration/adhesion [1,2,3,4]. Attempts have been made to reproduce these complex architectural and material properties in bioceramics by incorporating small biomolecules (i.e., peptides, organic acids, or molecular moieties) as additives [5,6,7,8,9,10,11,12,13,14]. With few notable exceptions, cement additives [9,10], which act by regulating the surface-driven processes of crystal nucleation, crystal growth, and cell-mediated dissolution/resorption [1,3,15,16], can only modestly improve the mechanical strength, microstructural organization, and rate of tissue integration of bioceramics [17].

We have discovered that a simple additive, phosphoserine, can recreate complex architectural and material properties in alpha tricalcium phosphate (αTCP) cements, reminiscent of native calcified tissues, and can produce entirely novel properties: strong adhesion to tissue and varied biomaterial surfaces, enhanced handling and setting, templated mineralization of nanoscale amorphous calcium phosphate, and stabilization of metastable ceramic phases. There have been only two prior reports of adhesive cements in the field of calcium phosphate bioceramics [9,10]. In both cases, the mechanisms underlying adhesion, and the methods by which this phenomenon can be extended to different calcium phosphates or other materials, are unclear. In this study we have produced phosphoserine-modified calcium phosphate cements (PM-CPC) by combining αTCP, a metastable ceramic that transforms into hydroxyapatite over the course of days to weeks in biological fluids [18], with phosphoserine. PM-CPCs hardened within minutes, rather than the days needed for the unmodified αTCP cement to cure, to form an organic/inorganic composite of amorphous, nanoscale calcium phosphate and phosphoserine, and αTCP.

We have evaluated whether PM-CPC meets the requirements for clinical use, as a calcified tissue adhesive, via mechanical tests on fresh tissues, with test samples being cured under wet-field conditions [19]. Chemical and structural analysis, with X-ray diffraction and electron microscopy, revealed that phosphorylated amino acids (i.e., phosphoserine) accelerated the setting, improved the mechanical strength, and enhanced the handling properties by (a) forming a macroscale organic phase that covers the surface of αTCP particles, thereby preventing the dissolution or reorganization of αTCP, and (b) facilitating the nucleation of nanoscale, amorphous calcium phosphate to produce a nanoscale organic/inorganic composite material. Phosphoserine was selected for this study because it is predominantly found in phosphoproteins that are involved in a wide range of biological processes; from adhesion, in marine “bioglues” [20,21,22], tissue adhesion, cohesion, and load dissipation in animals, to biomineralization, via matrix proteins [1,3,23], and matrix vesicles [2]. We hypothesized that some of these interesting biological and chemical properties might be recreated, to a limited extent, by incorporating phosphoserine into CPCs [8]. Herein, we report on the first class of bioinspired bioceramics that combine simple phosphorylated amino acids with tricalcium phosphate, to create novel, complex material properties.

## 2. Materials and Methods

### 2.1. Materials

All materials were purchased from Sigma-Aldrich (AB Sigma-Aldrich Sweden, Stockholm, Sweden), unless otherwise indicated. Monocalcium phosphate monohydrate (MCPM, Ca(H_2_PO_4_)_2_·H_2_O) was purchased from Scharlau (Barcelona, Spain). Alpha tricalcium phosphate (αTCP, Ca_3_(PO_4_)_2_) was synthesized by heating (Carbolyte oven CWF1300, AB Ninolab, Stockholm, Sweden) calcium carbonate and monocalcium phosphate anhydrous, at a 2:1 molar ratio, on a zirconia setter plate for 12 h, at 1450 °C. After quenching in air, the αTCP powder was dry milled (Reitsch PM400, AB Ninolab, Stockholm, Sweden) in a 500 mL zirconia milling jar, at 300 RPM for 15 min, with 100 grams of powder per 100 zirconia milling balls (10 mm diameter), and the purity was determined to be 98% wt% αTCP, with 2% wt% βTCP as an impurity, by Rietveld refinement of X-ray diffraction patterns (Profex software, PDF# 04-010-4348 αTCP, #01-074-0565 hydroxyapatite, #04-008-8714 βTCP, #04-007-9734 CaO) [24]. O-phospho-l-serine, hereafter referred to as phosphoserine, (>95%, Flamma SpA, Bergamo, Italy) was used as received. Fresh adult bovine and porcine tissue was obtained from Lövsta kött (Uppsala, Sweden), with approval from the Department of Agriculture. Cortical tissue from the bovine humeral diaphyseal shaft was cut into cubes (1 cm^3^) and stored in phosphate buffered saline at −20 °C. The cut surfaces were not polished or treated further. Tendon and cartilage samples were obtained from the bovine knee and hip joints, respectively, and stored as described for bone. Porcine skin (freshly harvested adult ears), heart, and liver were cut into strips (1 cm × 2 cm) and frozen for later use. Collagen films were generously provided by Viscofan GmbH (Weinheim, Germany).

### 2.2. PM-CPC Fabrication

Adhesive cements (PM-CPC) were created by premixing powders with defined molar ratios (mol%) of phosphoserine to αTCP. Unless otherwise indicated, all PM-CPC (%) compositions refer to mol%. Ultrapure water (18.2 Ω) was added at predetermined liquid-to-powder ratios (L/Ps), and samples were mixed with a spatula for 20 s. Scanning electron microscope (SEM) samples included: thin layer fracture samples (0.25 g cured on metal (steel, aluminum) or bone surfaces), and discs (2.0 g, cured in silicon molds, 8 mm diameter by 4 mm thick). A liquid-to-powder ratio of 0.25 g·mL^−1^ was used for SEM, compression, and adhesive testing samples, unless otherwise indicated. Samples were cured at 37 °C, in 100% relative humidity for compression or in ultrapure water for adhesive samples, for 24 h. Though PM-CPCs cured equally well in phosphate buffered saline (PBS), producing comparable adhesive shear strength, water was selected for testing to avoid confounding effects due to reprecipitation or dissolution caused by phosphate.

### 2.3. Mechanical Testing, Characterization

Samples for the adhesive shear test (bone or steel cubes, 1 cm^2^ contact area) were prepared by placing a thin layer (0.25 g) of PM-CPC on the surface, clamping the cortical bovine bone cubes together with universal grips, and submerging into water (37 °C, 24 h, calcified tissues) or 100% humid sealed containers (37 °C, 1–4 h, soft tissues). Samples were loaded to failure, on a Shimadzu AGS-X mechanical testing machine (Shimadzu Europa Gmbh, Duisburg, Germany). Lap shear samples were prepared by cutting the tissue to strips 1 cm × 2 cm × 0.5 cm thick, with a 1 cm^2^ adhered surface. Samples were tested at a crosshead speed of 1 mm per minute (compression and shear, calcified tissues) or 10 mm per minute (shear, soft tissues). Note that for collagen and liver, tissue samples failed cohesively before the PM-CPC bond (PM-CPC adhesive strength exceeded the cohesive strength of the material). Compression samples (cylinders 6 mm × 12 mm, cured as for shear testing) were tested at a crosshead speed of 1 mm per minute, with a spherically seated platen (Shimadzu #346-50639-32, Shimadzu Europa Gmbh, Duisburg, Germany). Prior to testing, cylinder surfaces were polished to 1200 grit fineness with silicon carbide polishing paper (Struers A/S, Bromma, Denmark). Cohesion was demonstrated with a Gilmore needle (#H-3150, Humboldt, IL, USA), using a 113 g load (needle tip diameter 2.12 mm) on unmodified CPC, and a 453 g load (needle tip diameter 1.06 mm) on PM-CPC. X-ray diffractograms (XRD) of powder samples were obtained on a Bruker D800 advance (Bruker Daltonics Scandinavia AB, Solna, Sweden), from 3 to 60 degrees, with a step size of 0.03 degrees per step, and a dwell time of 0.16 s.

### 2.4. SEM Analysis

SEM images were obtained on a Merlin field emission SEM (AB Carl Zeiss, Stockholm, Sweden), with an secondary electron in-lens detector, an acceleration voltage of 3 keV, and 195 pA current, for adhesive samples, at a working distance of 5 mm. Prior to SEM analysis, samples were sputtered with a (10 nm thick) coating of gold and palladium (Emitech SC7640, Quorum technologies, Kent, UK), at 2 kV for 40 s.

### 2.5. Solid-State NMR

All magic-angle-spinning (MAS) nuclear magnetic resonance (NMR) experimentation utilized a Bruker Avance-III spectrometer and a magnetic field of 9.4 T, which gives Larmor frequencies of −400.1 MHz and −162.0 MHz for ^1^H and ^31^P, respectively. Cement powders and polycrystalline *O*-phospho-l-Serine and l-serine samples were packed in 2.5 mm zirconia rotors and spun at 34.00 kHz. ^1^H and ^31^P chemical shifts are quoted relative to neat tetramethylsilane (TMS) and 85% H_3_PO_4_ (aq), respectively. ^1^H MAS NMR spectra were recorded by single pulses operating at a ^1^H nutation frequency ν_H_ = 80 kHz and using 4 s relaxation delays.

^1^H{^31^P} dipolar-mediated heteronuclear multiple-quantum coherence (D-HMQC) NMR [25] data were collected from CPCs incorporating 16 mol% of either phosphoserine or serine. ^1^H–^31^P multiple-quantum coherences were generated by one completed $SR4_{1}^{2}$ pulse sequence [26] applied to the protons, which corresponded to an excitation interval of 176 µs. Note that spin-diffusion due to ^1^H–^1^H interactions are suppressed by the $SR4_{1}^{2}$ sequence [26]. The dipolar recoupling pulses operated at ν_H_ = 68 kHz, whereas all strong 90°/180° ^1^H pulses of the D-HMQC scheme employed ν_H_ ~108 kHz, whereas 83 kHz was used for the ^31^P 90° pulse. For each real/imaginary data-set of a States-TPPI acquisition [27], typically 20(*t*_1_) × 600(*t*_2_) time-points were recorded and zero-filled to 128(*t*_1_) × 4096(*t*_2_) points, prior to 2D Fourier transformation. Relaxation delays of 2 s and dwell-times of Δ*t*_2_ = 7.20 μs were used together with Δ*t*_1_ = 117.6 µs and Δ*t*_1_ = 176.5 µs for the experiments with cements incorporating serine and phosphoserine, respectively, whereas a corresponding number of 1536 and 512 accumulated signal transients were recorded for each *t*_1_-value. The larger number of transients used for the serine-bearing cement reflects the lower affinity of serine (as compared with phosphoserine) to bind to the calcium phosphate component (see Section 3.5).

### 2.6. Statistics

Statistically significance differences were identified with SPSS software (version 22), with 1-way ANOVA, using Games-Howell post hoc analysis.

## 3. Results and Discussion

### 3.1. PM-CPC Setting and Compressive Strength

PM-CPCs were created by premixing phosphoserine and αTCP powders, with ultrapure water as the liquid (Figure 1a). Phosphoserine radically accelerated the setting kinetics of αTCP cement; even small amounts (<5%) of phosphoserine caused αTCP cement to harden and set within minutes, while unmodified αTCP cement required hours to set, and days to convert to hydroxyapatite. When the setting and cement cohesion were qualitatively evaluated with a Gilmore needle, PM-CPC easily supported a 453 g Gilmore needle within 10 min, while αTCP cement lost cohesion under the force of the lighter, 113 g Gilmore needle (Figure 1b). The compressive strength of the αTCP cement lacking phosphoserine, prepared under identical conditions as PM-CPC, reached 4.0 MPa, while PM-CPC reached 12-fold higher strengths (average of 51.4 MPa, 0% vs. 32%, Figure 1c). The L/P ratio of 0.25 was used to ensure all compositions were workable.

### 3.2. PM-CPC Adhesive Strength and Optimal Formulation

The accelerated setting reaction also produced rapid, strong adhesive bonding between PM-CPCs and calcified tissues, as demonstrated visually in Figure 2a, where bovine cortical bone cubes that were glued together with PM-CPC and allowed to cure for 20 min, in liquid, were able to support a 4 kg weight (4 kg/cm^2^ or 0.4 MPa). Though most adhesives require polished, etched, and level surfaces to bond properly, in the present study, all adhesive testing was conducted under the most challenging conditions: untreated and unpolished tissue surfaces, and cured in liquid, to approximate actual clinical conditions. Shear testing was performed with the test rig shown in Figure 2b, after cortical bovine bone cubes were glued together with PM-CPC and cured in liquid for 24 h. αTCP cements lacking phosphoserine are reported to produce 0.05 MPa of adhesive shear force [10]. In comparison, PM-CPC reached as much as 30-fold higher shear strengths (average of 1.8 MPa, 32 mol% PM-CPC, L/P 0.25, Figure 2c). Even stronger adhesive and compressive strengths were obtained with lower liquid-to-powder ratios, reaching a maximum strength of 4.8 MPa in shear (average 2.5 MPa), and 110.8 MPa in compression (average 103.5 MPa) (Figure 2d) at a L/P ratio of 0.15 mL/g. The optimal PM-CPC formulation, based upon the compressive and shear strength results, contained 20–50 mol% phosphoserine (Figure 1c, Figure 2c). Adhesiveness, particularly to wet and soft tissue surfaces, was a completely novel and unexpected property for a material that was mostly ceramic (80 wt% ceramic). The United States FDA has not approved a material for internal use as a bone tissue adhesive, therefore it is unclear which material is an appropriate standard [19]. Of the tissue adhesives reported in the scientific literature, PM-CPC outperformed other adhesives that had been proposed for similar applications. PM-CPC bonded to cortical bone up to 4-fold (4 MPa vs. 1 MPa, dry), and 10-fold stronger (4 MPa vs. 0.1–0.3 MPa, wet), under shear, than cyanoacrylates and other natural adhesives [28,29], respectively, and up to 100-fold stronger than fibrin glue (0.04 MPa) [30].

### 3.3. PM-CPC Adhesion to Soft Tissues and Biomaterials

In addition to the strong adhesion to calcified tissues, PM-CPCs also bonded to a wide range of other tissue surfaces, including soft tissues, internal organs, synthetic (steel)-, and naturally-derived (collagen sheets) biomaterials (Figure 3a–d). The failure mode was brittle, which is expected for a ceramic material, with sharp failure occurring cohesively rather than adhesively (Figure 3e).

### 3.4. PM-CPC Physiochemical Analysis of PM-CPC

The adhesive and compressive strength of PM-CPC varied based upon the mole percentage of phosphoserine. The fracture surface of discs composed of αTCP CPC, or varied PM-CPC compositions, were compared with scanning electron microscopy (SEM) (Figure 4a–f) to identify how phosphoserine affected the microstructure and compressive strength. In αTCP CPC, the αTCP remains as coarse, unreacted granular particles, with reprecipitated nano-hydroxyapatite plate and needle like crystals covering the particle surface (Figure 4a,c, 0 mol% PM-CPC, L/P 0.25), after 24 h. In contrast, in phosphoserine-doped CPCs (2 mol% PM-CPC, and 54 mol% PM-CPC, L/P 0.25), phosphoserine covered the granular surface of αTCP, and formed an organic interlayer (Figure 4b,e, 2 mol% PM-CPC, L/P 0.25). At low amounts of phosphoserine, where the compressive strength is also low, the interlayer appears to subsume nanohydroxyapatite; small plate-like crystals can be seen within the interlayer. At higher concentrations of phosphoserine, where the compressive strength is optimal (Figure 4c,f, 53 mol% PM-CPC, L/P 0.25), αTCP particles are no longer visible and the entire surface appears amorphous. Lower magnification SEM images clearly show the evolution from a granular to amorphous macrostructure, with increasing phosphoserine (Figure 4a–c), while higher magnification images of a fracture surface revealed an underlying porous organic architecture, and an amorphous surface covered with nanoscale mineral (Figure 4f). The cracks that appear on the surface in Figure 4c,f resulted from the fracture and lyophilization process, during the preparation for SEM imaging.

In αTCP CPC cured for 24 h, crystalline peaks for αTCP and hydroxyapatite (black circle and orange square, respectively, Figure 5a) are present in the X-ray diffractograms. As the concentration of phosphoserine increased, from 0% to 15% PM-CPC, hydroxyapatite peaks disappeared. In the range of PM-CPC formulations where the compressive strength is optimized (16% to 52% PM-CPC), crystalline peaks for both phosphoserine and hydroxyapatite were absent; only crystalline αTCP peaks, and a significant amorphous region seen in 10–20° 2-theta range, are visible. It should be noted that phosphoserine appears to recrystallize, likely due to supersaturation, at very high concentrations (80% PM-CPC). Collectively, the XRD data of PM-CPCs, cured for 24 h, suggests that the adhesive and cohesive strength decreases concurrently with a transition in the αTCP phase from crystalline to predominantly amorphous (>53% PM-CPC).

Amorphous calcium phosphates are unstable, and they readily convert to more stable phases in liquid. Since PM-CPCs are largely amorphous, by XRD analysis at higher concentrations of phosphoserine, we also evaluated whether PM-CPC remained amorphous in liquids or converted to more stable phases over time. After 14 days of curing, in humid (Figure 5b) or wet conditions (Figure 5c), αTCP CPCs converted into more stable hydroxyapatite, with more efficient conversion occurring in wet conditions. However, PM-CPCs with more than 2 mol% phosphoserine retained αTCP in crystalline form and prevented the development of crystalline hydroxyapatite, even after 14 days in liquid (Figure 5c).

### 3.5. The Organic/Inorganic Interface

Solid-state NMR spectroscopy was employed to gain structural insight across a sub-nanometer-length scale, using D-HMQC [25] ^1^H{^31^P} NMR experiments to specifically probe the organic/inorganic *interface* of the PC-CPC samples. The 2D NMR spectra displayed in Figure 6 were recorded from cements incorporating 16 mol% of either phosphoserine or l-serine. The detected proton signals mainly stem from the organic molecules, as is evident from comparing the projections of the D-HMQC NMR spectra along the ^1^H dimension (shown on top of each 2D NMR spectrum) with their directly excited ^1^H NMR counterparts obtained from well-crystalline phosphoserine and serine. The broader ^1^H signals observed from the CPCs, relative to the well-crystalline phosphoserine/serine powders, reflect the structural disorder of the organic polymeric networks in each cement matrix.

The amorphous nature of *both* the organic and inorganic components that interface each other was evidenced by the broad signals observed in the projection along the vertical ^31^P dimension, which solely reveals chemical shifts associated with *ortho*phosphate groups. The slightly differing peak maxima along the ^31^P projection of the D-HMQC spectra from cements incorporating phosphoserine (~1.4 ppm) and serine (~2.5 ppm) originated from contributions of NMR signals from both organic and inorganic phosphate groups in the case of phosphoserine (Figure 6a), whereas all resonances stemmed from inorganic phosphates in the serine-bearing cement (Figure 6b). Notably, the peak maximum ~2.5 ppm of the latter is close to the shift ~3 ppm typically observed from amorphous calcium phosphate (ACP), which is known to also be present as a surface layer in both synthetic and biogenic apatites [31,32].

Altogether, these results demonstrate that *the two amorphous organic/inorganic components of the cement are intimately bonded at a molecular scale*, presumably mainly by electrostatic interactions and hydrogen bonds. We comment that ^31^P MAS NMR spectra recorded directly by single pulses (not shown) reveal a set of sharp peaks from unreacted αTCP, besides the dominating broad resonances from the amorphous organic/inorganic components. Yet, no resonances from α-TCP are detected in the HMQC 2D NMR spectra, due to the absence of protons in its structure and the much longer (>1 nm) distances between the αTCP crystallites and the proton-bearing organic/ACP components, in full consistency with the SEM results in Figure 4.

### 3.6. Adhesive Mechanism and Macrostructure of PM-CPC

SEM analysis of small adhesive samples revealed that adhesion occurs via a distinct organic phase that wetted the substrate surface (Figure 7a). This organic/amorphous surface layer becomes mineralized as PM-CPC sets and hardens, thereby “cementing” the adhesive interface. Cross-sections of PM-CPC discs confirmed that the outer surface was morphologically amorphous/organic (as if the surface were coated with protein or polymer), and heavily mineralized with nanoscale amorphous calcium phosphate spheres (Figure 7b). The fracture surface of thin layer PM-CPC, seen after shear testing, confirmed that the outer organic layer is connected to a dense mineral interior that includes αTCP particles, via a highly porous, organic network (Figure 7c). The honeycomb-like organic network was composed of mesoscale vesicles (red arrow, Figure 7c), upon which nanoscale calcium phosphate nucleated. In contrast, αTCP cement without phosphoserine (0% PM-CPC) developed random aggregations of nanohydroxyapatite, with very different macrostructures (Figure 7d). The excellent cohesive properties (compressive strength) of PM-CPC likely arose from mineral nucleation onto the entanglements within the organic network, thereby creating “mineralized bridges” (Figure 7c) at network intersections and effectively crosslinking the organic network. The organic network created macroscale disorder in the final ceramic, similar to what occurs naturally during biomineralization of amorphous mineral, and during the crystallization of biogenic ceramics [33,34]. Thus, PM-CPCs appear to be a novel, bioinspired, nanoscale composite of organic and inorganic phases, with what appears to be a form of hierarchical organization.

### 3.7. Hierarchical Organization and Templating of PM-CPC by Phosphoserine

Based upon the present observations, we propose a self-assembly process that creates the unique microstructure of PM-CPCs (Figure 8). PM-CPCs occur when phosphoserine and calcium phosphate (αTCP) are combined in a supersaturated solution. Phosphoserine initially dissolves during the PM-CPC setting reaction, but in the supersaturated conditions of a cementitious reaction (phosphoserine concentration exceeds 4 M, while solubility in water is <<1 M, supersaturated levels of calcium and phosphate also present from dissolving αTCP), phosphoserine is found in both hydrated (soluble fibrils, Figure 8a) and partially hydrated forms (self-assembled into mesoscale particles or molecular clusters to minimize hydration, Figure 8b) [35,36,37]. Both forms of phosphoserine contribute to the observed organic network (Figure 8c), where phosphoserine acts as a nucleation initiator for amorphous calcium phosphate. In Figure 8b,c, the nanoscale mineral appears white, while the organic matrix appears grey. Mesoscale phosphoserine particles could display a hydrophilic surface and hydrophobic interior, or vice versa, as occurs with other self-assembled organic structures (i.e., micelles). In the presence of a charged αTCP particle surface, during the setting reaction, it is likely that the charged, hydrophilic portion of phosphoserine particles (white spheres, Figure 8b, top) covers the αTCP grain, thereby attenuating interactions with surrounding aqueous liquids, which explains how the αTCP phase remains crystalline in PM-CPCs for weeks, even in liquids. The aggregation and subsequent mineralization of the organic network within PM-CPCs produce a hierarchical macrostructure. During adhesion, the organic surface layer of PM-CPC initially wets an adherend substrate surface (Figure 8c), before subsequently mineralizing during the cementitous setting reaction. Immediately beneath the PM-CPC surface, the organic phase also entangles with, and connects to, the denser inorganic, mineral interior (Figure 8d). The macroscale disorder created by the organic network, and the amorphous nature of the nanoscale calcium phosphate that nucleates onto the network, contribute to the amorphous spectra seen with X-ray diffraction.

Adhesion may arise from nanoscale-charged surfaces [38,39,40,41], hydrogen bonding [42], and hydrophobic/hydrophilic surface wetting, which commonly occur with amino acids and peptides [43]. PM-CPCs are nanoscale composites, which include mesoscale organic, and nanoscale inorganic (CPC), particles. We propose that the high surface energy and charge density associated with nanoscale calcium phosphate [38,39,40,41], and the charged organic layer on the surface of PM-CPC, facilitate the initial wetting of a substrate. Subsequent mineralization of the organic surface/interface then effectively “cements” the adsorbed organic layer to substrate surfaces, thereby partially creating adhesion. As the organic network within PM-CPC mineralizes, effectively creating crosslinking, the cohesive strength increases as well.

## 4. Conclusions

In this study we have demonstrated, for the first time, that phosphoserine transforms αTCP cements into strong tissue adhesives that rapidly cure, even in wet-field conditions. PM-CPCs can bond to a diverse array of surfaces, including biomaterials (i.e., metals, polymers), and calcified and soft tissues. This novel property arises from the unique nano-scale interactions between phosphoserine and amorphous calcium phosphate, and the hierarchical organization of the organic phase in PM-CPCs. PM-CPCs combine useful properties from both the organic and inorganic substituents: they exhibit strong tissue adhesive strength, despite being largely ceramic (60–80 wt% ceramic). Simultaneously, PM-CPCs with large amounts of organic substituent (phosphoserine, 53% PM-CPC) still produce compressive strengths that match the native αTCP CPC, and that set/cure within hours rather than days.

An adhesive with the ability to bond calcified tissues and withstand physiological tensile, shear, and compressive loading would be appropriate for many clinical applications, including joint reconstruction (e.g., elbow), osteochondral reconstruction (e.g., articular cartilage in the knee), and for the reconstruction of catastrophic tissue injuries [19,44]. An adhesive that bonds both soft and hard tissues could also redress injuries at the hard/soft tissue interface (e.g., tendon to bone). PM-CPC appears to be a promising biomaterial, suited for numerous clinical applications, that requires further evaluation as both a tissue adhesive, and a bioceramic. Nevertheless, prior to clinical trials, it is necessary to document the safety and efficacy during experimental in vitro and in vivo conditions. Such measures include the determination of the PM-CPC degradation rate, cyto-, and biocompatibility, temporal changes of adhesive/mechanical strength, and any pro-regenerative effects at the application site.

## Figures and Tables

**Figure 1 materials-11-02492-f001:**
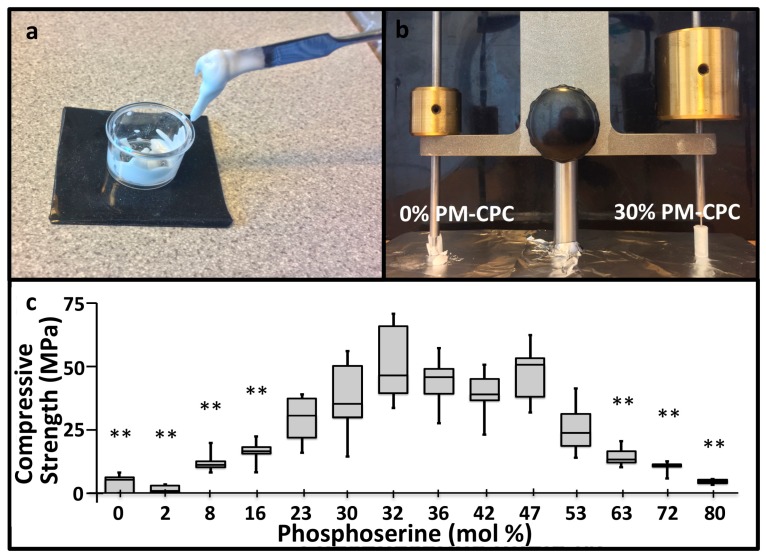
Physical appearance, setting, and compressive strength of phosphoserine-modified calcium phosphate cements (PM-CPCs). (**a**) PM-CPCs are liquids, which are easy to handle and mix, unlike αTCP cements, which are granular, chalky, and challenging to deliver. (**b**) The rapid cohesive strength of PM-CPC is visually represented (0% PM-CPC vs. 30% PM-CPC, 0.25 liquid-to-powder ratio (L/P)) by placing the Gilmore needle after curing/setting for 10 min. (**c**) The compressive strength of PM-CPCs with varied phosphoserine concentration (0.25 L/P, box plots represent the minimum, maximum, and median values, with whiskers denoting the first and third quartiles (25%, 75%)).Statistical analysis compared each group to the 30% PM-CPC using ANOVA with Games-Howell post hoc analysis, with * indicating *p*-values below 0.05, and ** indicating *p*-value below 0.01.

**Figure 2 materials-11-02492-f002:**
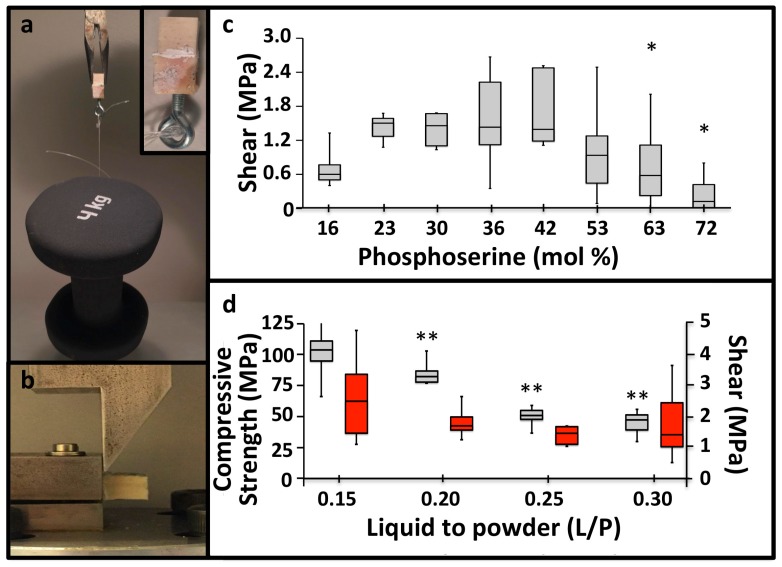
PM-CPC bonds to calcified tissues. (**a**) The adhesive strength of PM-CPC to calcified tissues is visually represented by a 1 cm^2^ cortical bone surface, bonded with PM-CPC, supporting 4 kg (0.4 MPa) after curing/setting for 20 min (inset is an expanded image of the glued bone cube). (**b**) The mechanical test rig used for shear testing. (**c**) The shear strength of cortical bone cubes glued together with varied compositions of PM-CPC for 24 h, in liquid. (**d**) The shear and compressive strength of 30% PM-CPC with varied L/P ratios (Figure 2d, 0.15 to 0.3 L/P) (compression—grey bars; shear—red bars; differences in shear strength, between liquid-to-powder ratios compared to 0.15 L/P, are indicated with * (*p* < 0.05), and ** (*p* < 0.01), ANOVA with Games-Howell post-hoc). Box plots represent the minimum, maximum and median value, with whiskers denoting the first and third quartiles (25%, 75%). In Figure 2c, statistical analysis was used to compare each group to the 30% PM-CPC using ANOVA with Games-Howell post hoc analysis, with * indicating *p*-values < 0.05. In Figure 2d, the compression and shear samples were analyzed, separately, with Games-Howell, with * indicating *p*-values below 0.05, and ** indicating a *p*-value below 0.01.

**Figure 3 materials-11-02492-f003:**
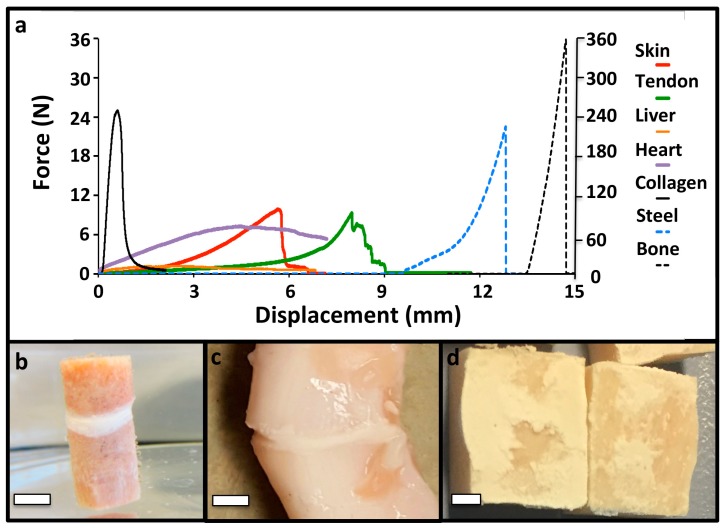
PM-CPC bonds to different tissue types and biomaterials. (**a**) Representative force/displacement curves of PM-CPC bonded to different tissues and biomaterials (bone/steel use the right y-axis (Force); all other tissues/materials use the left y-axis (Force)). Note that the initial displacement values have been shifted for clarity, while the total displacement remains accurate, for each sample (i.e., steel samples were displaced approximately 3 mm before failure, though in (**a**) the start of displacement has been shifted by 9.5 mm so it no longer overlaps with soft tissue samples). The test surface area was approximately 1 cm^2^. Soft tissues and collagen were tested by lap shear, as described in the Methods section. (**b**) Representative images of articular cartilage; (**c**) tendon; (**d**) or bone after shear testing to failure, glued to the same tissue type (i.e., articular cartilage to articular cartilage). Note the thin, white layer of PM-CPC between the two segments of tendon (**c**). The PM-CPC failure mode was cohesive, mixed failure (**d**). In (**b**–**d**) 30% PM-CPC was used. Scale bar dimensions are 5 mm, 3 mm, and 2.5 mm for (**b**), (**c**), and (**d**), respectively.

**Figure 4 materials-11-02492-f004:**
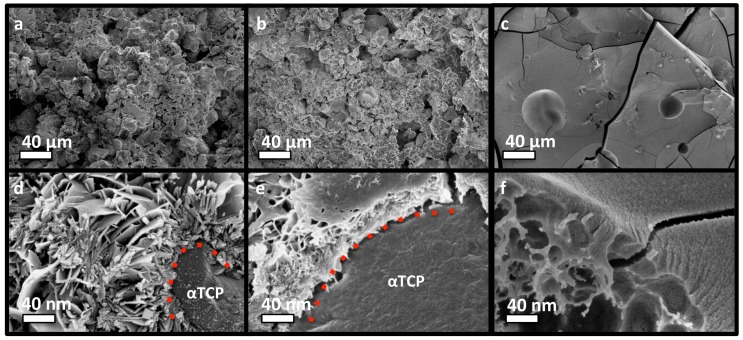
SEM images of the fracture surface of discs made of PM-CPCS. (**a**–**c**) The microstructure of PM-CPC changed as the mol% of phosphoserine increased from 0% (**a**) to 2% (**b**) to 53% (**c**); shifting from agglomerations of αTCP particles (**a**) to an amorphous solid (**c**). (**d**–**f**) Higher magnification images (100,000×), from 0% (**d**) to 2% (**e**) to 53% (**f**)) revealed that phosphoserine formed a thin organic layer that covered the outside of individual αTCP particles, and replaced, or subsumed, the nanohydroxyapatite that would normally have reprecipitated ((**d**) vs. (**e**)). The fracture surface also revealed that the underlying structure of PM-CPC was porous and organic, while the surface was composed of an amorphous/ organic layer that contains nanoscale mineral (**f**). Scale bar dimensions are indicated for each row; 40 µm (**a**–**c**), 40 nm (**d**–**f**).

**Figure 5 materials-11-02492-f005:**
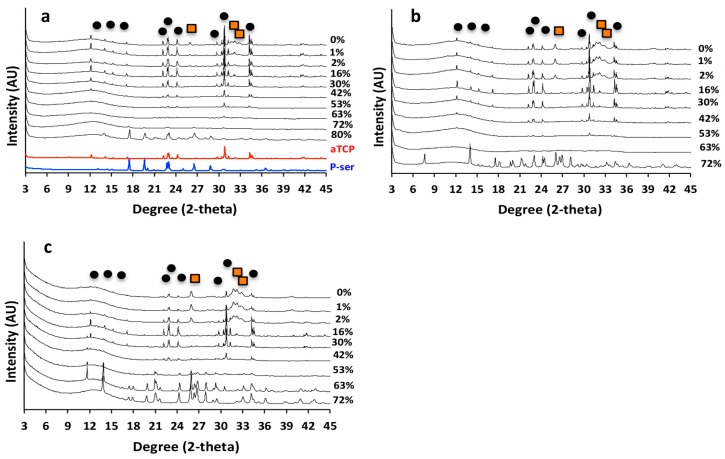
(**a**,**b**,**c**) X-ray diffractograms of PM-CPC after curing for 1 (a) or 14 (b) days in 100% humidity, or 14 days in water (c). Crystalline hydroxyapatite peaks (orange squares, a) appeared within 24 h in αTCP CPCs (0% PM-CPC), while as little as 2–16% phosphoserine inhibited hydroxyapatite formation and inhibited the dissolution and transformation of αTCP (black circles, a), even after curing for 14 days in liquid (**c**). Crystalline hydroxyapatite peaks are indicated by orange squares (PDF# 01-074-0565), while crystalline αTCP peaks are indicated by black circles (PDF# 04-010-4348).

**Figure 6 materials-11-02492-f006:**
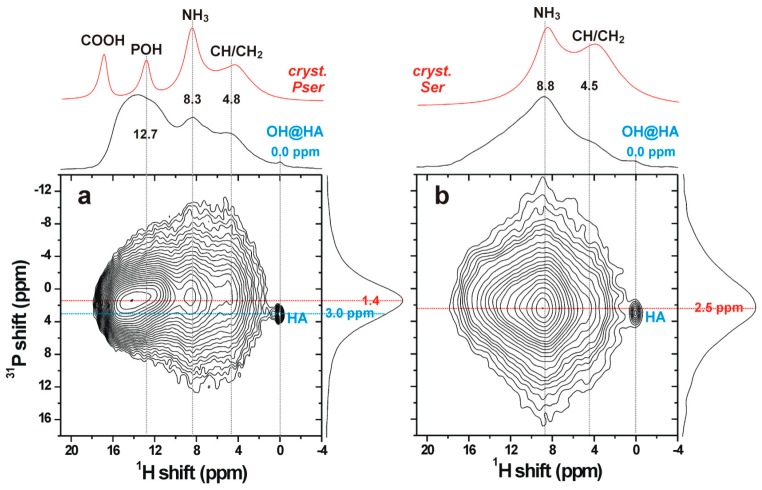
(**a**,**b**) ^1^H{^31^P} D-HMQC NMR spectra recorded from cements incorporating 16 mol% of (**a**) O-phospho-L-Serine and (**b**) L-serine. NMR signals are only detected from ^1^H and ^31^P nuclei in close spatial proximity (<0.5 nm), where each proton resonance (at chemical shift δ_H_), appearing along the horizontal dimension of the 2D NMR spectrum, is correlated with its respective ^31^P NMR signal (at chemical shift δ_P_) along the vertical spectral dimension. Projections along the ^1^H and ^31^P dimensions are shown at the top and to the right of each 2D NMR spectrum, respectively, whereas the topmost red traces represent ^1^H magic-angle-spinning (MAS) NMR spectra recorded directly by single pulses. The 2D NMR spectra from both the phosphoserine and serine cements confirm that all detected signals originate from amorphous organic/inorganic components, except for a minute (<1% of the total integrated intensity) and narrow peak (labeled “HA”) at the shift-coordinate {δ_P_, δ_H_} = {3.0, 0.0} ppm, which stems from well-ordered hydroxyapatite.

**Figure 7 materials-11-02492-f007:**
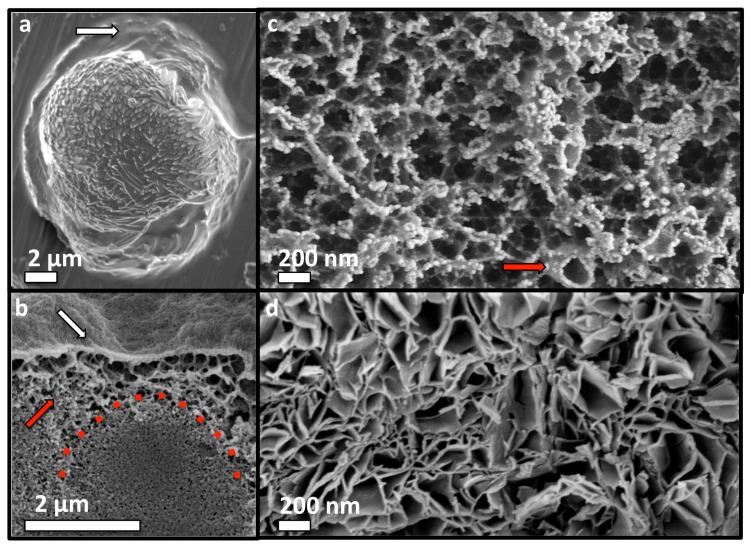
Adhesive mechanism and macrostructure of PM-CPC. (**a**) Adhesion and wetting of a substrate surface occurred via an organic phase that appeared on the surface of PM-CPC (white arrow, 72% PM-CPC, aluminum surface). (**b**) A cross-section of a PM-CPC disc confirmed that the surface appeared amorphous/organic (white arrow, 16% PM-CPC, 0.25 L/P), though closer inspection revealed that the organic phase was heavily mineralized with small, spherical clusters of calcium phosphate. The organic phase extended from the surface to the interior of PM-CPC, in the form of a porous, honeycomb like network ((**b**), red arrow). Unreacted αTCP particles appear in the interior of PM-CPC (red dotted line). (**c**) A cross-section of the adhesive interface (53% PM-CPC, 0.25 /LP), taken at the fracture interface of thin layer PM-CPC, after shear testing, revealed the organic network seen in (**b**), is comprised of organic (phosphoserine) meso-scale particles, which form an interconnected fibrillar, polymer like network. Individual meso-scale organic particles that comprise the network can be distinguished ((**c**) red arrow). (**d**) In comparison, αTCP cement without phosphoserine (0% PM-CPC, 0.25 /LP) contains irregular porosity arising from the random aggregation of plate-like hydroxyapatite crystals. Scale bar dimensions are 2 µm in (**a**,**b**); and 200 nm in (**c**,**d**).

**Figure 8 materials-11-02492-f008:**
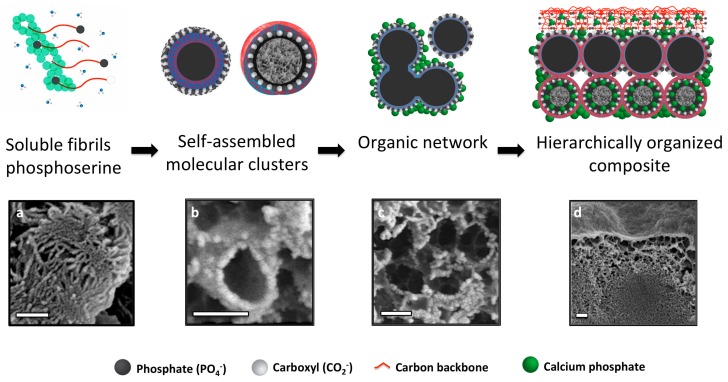
Hierarchical organization and templating of PM-CPC by phosphoserine. The proposed self-assembly process (top row images) is supported with SEM images from PM-CPC samples (bottom row images). (**a**–**c**) Phosphoserine was observed in two forms: (**a**) fibrillar (thin layer fracture surface, 2% PM-CPC) and (**b**) as meso-scale particles (thin layer fracture surface, 53 mol% PM-CPC). (**c**) In cementitious reactions phosphoserine self assembles into meso-scale particles, and subsequently aggregates into an organic network (thin layer fracture surface, 53 mol% PM-CPC). The charged moieties (phosphate and carboxylate) of phosphoserine serve as nucleation points, which template the formation of nanoscale amorphous calcium phosphate spheres, seen individually in (**b**) and covering the fracture interface in (**c**). (**d**) The organic network integrates meso-scale organic and nanoscale inorganic phases, into a composite material that appears hierarchically organized (16 mol% PM-CPC). Scale bar: 200 nm for all images.

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
