# Peer review of "A Novel Class of Injectable Bioceramics That Glue Tissues and Biomaterials"

_materials, 2018, doi:10.3390/ma11122492_

Reviewer 1 Report

The manuscript is nice to be read cause the topic is well structured into the text. 

The research is quite actual and authors made exccellent job on presenting thei results about this novel class of injectable bioceramics material. 

However introduction section and discussion section should be enlarged. In the introduction section, in order to have a more attractive paper, the clinical rationale of the application of this new devices should be added, adding a paragraph talking about the clinical evidence or possible clinical application. 

I found the figure 8 really impressive and clear. It needs to be expanded this concept maybe adding some recent papers useful for the discussion secion.

Some suggestions 

Facial Bone Reconstruction Using both Marine or Non-Marine Bone Substitutes: Evaluation of Current Outcomes in a Systematic Literature Review.

Cicciù M, et al. Mar Drugs. 2018 Jan 13;16(1). pii: E27. doi: 10.3390/md16010027. Review.

Author Response

Reviewer # 1 comment: “in order to have a more attractive paper, the clinical rationale of the application of this new devices should be added”

 Reply and action: We agree with the Reviewer. The following text has been added to the Conclusion section, and the reference recommended by the reviewer has been added to the References:

 Added text (in italics), Lines 430-436:

An adhesive with the ability to bond calcified tissues and withstand physiological tensile, shear and compressive loading would be appropriate for many clinical applications, including joint reconstruction (e.g. elbow), osteochondral reconstruction (e.g. articular cartilage in the knee), and for reconstruction of catastrophic tissue injuries [19,44]. An adhesive that bonds both soft and hard tissues could also redress injuries at the hard/soft tissue interface (e.g. tendon to bone). PM-CPC appears to be a promising biomaterial, suited for numerous clinical applications, that requires further evaluation as both a tissue adhesive, and a bioceramic. Nevertheless, prior to clinical trials, it is necessary to document the safety and efficacy during experimental in vitro and in vivo conditions. Such measures include the determination of the PM-CPC degradation rate, cyto- and biocompatibility, temporal changes of adhesive/mechanical strength and any pro-regenerative effects at the application site.

Citation 44 (added):

Cicciù, M., Cervino, G., Herford, A.S., Famà, F., Bramanti, E., Fiorillo, L., Lauritano, F., Sambataro, S., Troiano, G., Laino, L. Facial Bone Reconstruction Using both Marine or Non-Marine Bone Substitutes: Evaluation of Current Outcomes in a Systematic Literature Review. Marine Drugs 2018, 13, doi: 10.3390/md16010027.

Reviewer 2 Report

The authors prepared calcium phosphate bone cements based on a-TCP hydrolysis to calcium deficient hydroxyapatite in the presence of phosphoserine. They found that in te presence of phosphoserine the cements are transformed into strong tissue adhesives. This is a new information in the literature and the work is novel and the results are very interesting to scientist working with bone filler materials. The explanation regarding the mechanism of the adhesion is well explained and is based on experimental data. To my opinion the work has to be published after minor revision.

 Minor comments

1.    The authors prepared a-TCP by solid state reaction between MCPM and calcite at 1150 oC. This temperature is rather low to receive  pure a-TCP. Formation of b-TCP. is common in th eliterature. Did they observed any b-TCP in their solid ?

2.    Probably showing an EDS microanalysis (e.g. line scan for certain elements across the adhesion interface) should be interesting to people who are not very familiar with NMR analysis

3.    Line 248 legend: “..SEM and XRD analysis of …” There are no XRD graphs in this Figure

 Author Response

Reviewer #2 comments and response:
1. The authors prepared a-TCP by solid state reaction between MCPM and calcite at 1150 oC. This temperature is rather low to receive pure a-TCP. Formation of b-TCP. is common in th eliterature. Did they observed any b-TCP in their solid ?
Reply: The authors actually used a temperature of 1450oC, and this was an error in the text of the manuscript. The authors have corrected this error with the following text, and are willing to include an XRD spectrum, in the supplementary material section of the manuscript, if requested by the reviewer. bTCP was detected in the aTCP, as an impurity at 1-2% wt%. Since this low amount of impurity is within the error of the analytic method (Rietveld refinement, X-ray diffraction), it was not investigated further.
Action:

Added text (in italics), Lines 84-88
After quenching in air, the αTCP powder was dry milled (Reitsch PM400) in a 500 mL zirconia milling jar, at 300 RPM for 15 minutes, with 100 grams of powder per 100 zirconia milling balls (10 mm diameter), and the purity was determined to be 98% wt% αTCP, with 2% wt% βTCP as an impurity, by Rietveld refinement of x-ray diffraction patterns (Profex software, PDF# 04-010-4348 αTCP, #01-074-0565 hydroxyapatite, #04-008-8714 βTCP, #04-007-9734 CaO) [24].
2. Probably showing an EDS microanalysis (e.g. line scan for certain elements across the adhesion interface) should be interesting to people who are not very familiar with NMR analysis
Reply: The authors have attempted to use EDS to characterize the interface, however the highly organic nature of the composite resulted in rapid deterioration (burning in the electron beam) of the surface. This interferes with the data collection, as high beam voltage is needed to collect enough counts for an accurate reading. Furthermore, the presence of organic phosphate and inorganic calcium phosphate creates large uncertainty when attempting to evaluate the inorganic portion by EDS, as the resulting calcium to phosphate ratios vary significantly. It is difficult to determine whether the collected data reflects inherent stoichiometric variation arising from: the amorphous nature of the inorganic portion, the presence of organic phosphate, or interference from decomposition of the sample. A thorough analysis would require a complete separate study, effectively. Therefore, the authors have elected to summarize the results in the present manuscript without using EDS analysis.

 Action: No action.

3. Line 248 legend: “..SEM and XRD analysis of …” There are no XRD graphs in this Figure a
Reply and action: We thank the Reviewer for this observation. The XRD spectra are found in Figure 5. The text has been corrected accordingly. All references to XRD have been removed from line 248 (Figure 4).

Reviewer 3 Report

it was a pleasure to review this manuscript on the development of a novel phosphoserine modified calcium phosphate cement with enhanced properties. The manuscript is very well written and to my opinion presents a novel material. The data are enough to substantiate the conclusions and I believe it can be published following minor changes.

Comments: 

1. Please add statistics in Fig 1C and 2C

2. Line 187 please check the "..."

3. In figure 4 there are no XRD images. Please add them.

4. Please mention that extra in vivo results are necessary to judge the suitability for human application and suggest specific future experiments

Author Response

Reviewer #3 comments and response

1. Please add statistics in Fig 1C and 2C
Reply and actions: We thank the Reviewer for this comment. The following statistical analyses and appropriate text have now been added:
In methods section:

“Statistically significance differences were identified with SPSS software (version 22), with 1-way ANOVA. In Fig. 2d the compression and shear samples were analyzed, separately. Post-hoc analysis was conducted with Games-Howell, with * indicating p-values below 0.05, and ** indicating p-value below 0.01.”
In Fig. 1 legend:
“Statistical analysis compared each group to the 30% PM-CPC, indicated by the longer bracket, using Games-Howell post-hoc analysis, with * indicating p-values below 0.05, and ** indicating p-value below 0.01”
In Fig. 2 legend:
“In Fig. 2c statistical analysis compared each group to the 30% PM-CPC using Games-Howell post-hoc analysis, with * indicating p-values < 0.05.”
2. Line 187 please check the "..."
Reply and action:

The authors thank the reviewer for catching this error and have corrected the text with the liquid to powder ratio (L/P).
3. In figure 4 there are no XRD images. Please add them.
Reply and action:

The authors thank the reviewer for catching this error and have corrected the text by moving all XRD spectra to a new figure (Figure 5) and removing all reference text to XRD, from Figure 4.
4. Please mention that extra in vivo results are necessary to judge the suitability for human application and suggest specific future experimentsa
Reply and action:

The authors thank the reviewer for this helpful suggestion and have added the following new text in the Conclusion section:
Nevertheless, prior to clinical trials, it is necessary to document the safety and efficacy during experimental in vitro and in vivo conditions.  Such measures include the determination of PM-CPC material degradation rate, cyto- and biocompatibility, temporal changes of adhesive/mechanical strength and any pro-regenerative effects at the application site.